# The Role of Adequate Resources, Community and Supportive Leadership in Creating Engaged Academics

**DOI:** 10.3390/ijerph18052776

**Published:** 2021-03-09

**Authors:** Marit Christensen, Jeremy Dawson, Karina Nielsen

**Affiliations:** 1Department of Psychology, Norwegian University of Science and Technology, N-7491 Trondheim, Norway; k.m.nielsen@sheffield.ac.uk; 2Institute of Work Psychology, Sheffield University Management School, University of Sheffield, Sheffield S10 1FL, UK; J.F.Dawson@Sheffield.ac.uk

**Keywords:** work engagement, community, academic resources, leadership, academia

## Abstract

The vast majority of research in academia focuses on the adverse working conditions and poor wellbeing. The present paper presents a positive view on the factors that may promote work engagement in academia. Based on conservation of resources theory, we suggest that academic resources may be related to a social community at work, which in turn creates work engagement among academics. Having positive leadership in the form of fair leadership may be an important contextual factor ensuring that resources are shared fairly and openly. In a study of 1499 academics in Norwegian universities, we found that sufficient administrative resources to support teaching duties were positively related with work engagement, and that a sense of community mediated the relationship between academic resources for teaching and work engagement. These results propose that building academics’ social resources by providing them with the necessary resources to perform their jobs will buffer the impact of a leadership that is perceived to be unfair and help them to perform their work in a positive way. Our results carry important implications for how positive psychology may be used to support engaged workers in academia.

## 1. Introduction

Most current research on the situation in academia paints a bleak picture. Austerity has led to cuts in funding with universities across the globe moving to models of new public management, increased competition for students and research funding [1]. The consequences of these changes have been adverse working conditions, in the form of high teaching loads [2], poor work–life balance with research being carried out after official working hours [2], and decreased collegiality [3,4]. As a response to this bleak picture of life in academia, a call has been made to understand how these difficult contexts within universities are forced to operate can be mitigated [1] to identify the factors which may promote work engagement among academics. From a positive psychology perspective, it is important to identify the resources, which may not only minimise the negative impact on mental health, but also provide insights into how universities and their management can promote positive outcomes for academics such as work engagement [5,6]. Work engagement has been closely linked to performance and productivity [7,8]. In the present study, we explore the working conditions, which may have a positive impact on academics. Specifically, we explore the role of having adequate academic resources, a social community at work and a fair leadership in engaging academics in Norwegian academia.

### Hypothesis Development

We based our development of hypotheses on the conservation of resources theory (COR) [9]. The basic tenet of COR is that individuals strive to protect existing resources and acquire additional resources [9,10]. Resources are defined as ‘anything that helps individuals achieve their goals’ [11], in the present study, the outcome variable is work engagement. Work engagement was chosen as an outcome in this study because earlier research has shown that it is found to be closely related to academic productivity, i.e., teaching and research [8]. Work engagement is defined as ‘… a positive fulfilling work-related state of mind, that is characterized by vigour, dedication and absorption’ [12] (p. 74). Studies suggest that work engagement is of importance for the motivation process and for performance at work [8,13,14,15]. The COR theory contributes to understanding the function of job resources in combination with work engagement. The resources contribute to reducing the negative impact of work demands; they are functional in achieving goals and stimulate growth and personal development. This is explained by a gain spiral where a reciprocal causal relationship between the resources increase work engagement [16].

One of the key cuts in academia is administrative resources; academics are increasingly required to take on administrative tasks that were previously performed by specialised staff [17] Competition is rife in academia with academics fighting for resources and the higher administrative workload has taken away time from the delivery of high-quality teaching and research [1,18]. A survey among academics in Australia found that only 1% of respondents felt there was sufficient staff to get the work done effectively [3] and 44% reported doing work for which they were not compensated, both of which were found to be profound sources of dissatisfaction. Therefore, there is an urgent need to understand the influence of having sufficient academic resources on academics. Academics who perceive they get the necessary administrative support to fulfil their teaching duties are likely to be more engaged in their work [19]. As they do not need to spend time on what could be perceived as tasks that do not offer career opportunities or delivering high-quality teaching and research, they may be more engaged. One of the few studies to explore the role of resources in academia found that adequate resources for providing research and administrative support was positively related to work engagement [20]. To replicate these results, we formulated our first hypothesis:

**Hypothesis** **1** **(H1).**
*There is a positive relationship between academic resources and work engagement.*


As mentioned above administrative support is sparse in academic institutions [21]. Duffy argued that organizational climates characterized by competition as would be the case in academia when there is limited access to administrative resource, bullying is the result [22]. According to negative gain spirals of the COR theory [23], the explanation for this would be resource depletion. When academics feel under-resourced and they have to fight colleagues for resources, they may expend personal resources to gain other resources at the expense of other colleagues and may come to see colleagues as competition or the enemy threatening to take away their administrative resource. From a positive gain spiral perspective [23], where administrative resources are perceived to be adequate, then there is less reason for academics to compete for such resource and they are less likely to feel that colleagues threaten to take away this resource. Furthermore, the need for relatedness is a basic human need that individuals thrive to fulfil [24]. Therefore, it is likely that as academic do not see their colleagues as competition, they may seek to fulfil their need for relatedness through their colleagues. When administrative resources are perceived to be sufficient, this enables academics to engage in efforts to gain additional resources and fulfil their basic need for relatedness, which is also considered a central part of sense of community [25]. Central to sense of community is the feeling of being part of a great whole and the work unit having a positive atmosphere.

A cross-sectional study among Australian academics found collegiality to be positively related to commitment and propensity to remain and negatively associated with job stress [26]. Social resources more broadly have been significantly related to work engagement [27].

We thus formulated our second hypothesis:

**Hypothesis** **2** **(H2).**
*A social community at work mediates the relationship between academic resources and work engagement.*


A relatively unexamined element of COR theory is the concept of passageways [11]. Passageways are the contextual factors, which may accelerate the impact of resources, for better or for worse [10]. Taking a positive spin on passageways, these can be seen as broaden-and-build resources [28] that may help academics gain further resources, e.g., social resources.

Social information processing theory [29] suggests that individuals’ perceptions and behaviours are formed by cues they receive from the social context and such social cues may influence the academics’ ability to acquire additional resources. Gomes and Knowles point to the fact that there is not so much research on how leaders contribute to departmental culture, collaborative atmosphere, and departmental performance’ [30]. Creswell and colleagues found that a common feature between departmental leaders that had been nominated as excellent was that they establish a collective departmental vision or focus [31]. Previous research has found the research has found fairness and reasonable job responsibilities are related to sense of community [32] and, therefore, we would suggest that having adequate administrative resources which may be connected to having reasonable job responsibilities may be related to a sense of community.

Leaders are well-known as enablers of resources [33] and the behaviours of leaders may send important cues as to resource investment and may shape the manner in which academics use resources to gain additional resources [11].

In other words, leadership behaviours may act as important passageways moderating the positive resource gain spiral between academic resource and a social community at work. Although academic resources may be available, this does not guarantee that resources are fairly distributed [19].

Previous research has found that uneven workloads are prevalent in academia with those lower in the hierarchy take on tasks not directly linked to career progression and self-fulfilment [19], despite the attempts of workload allocation models to ensure equitable workloads [34]. Despite official Humboldtian principles, academic managers often allocate higher teaching workloads to less research active staff, enabling research-intensive staff to focus on research [35]. Such practices create a further divide between academics as some are thereby given opportunities to engage in career-promoting tasks while others are confined to less career-promoting tasks such as administrative or teaching tasks [36]. A leadership perceived by academics to be fair may make the association between academic resources and a social community at work stronger, as such leaders ensure that academic resources are distributed evenly among academic staff, i.e., they have the necessary support to perform both teaching and research duties. Perceptions of justice has been found to be an important signal of resource investment [37]. Therefore, a fair leadership may be an important passageway for administrative resources to facilitate a social community as it will be assumed that investing academic resources to promote a social community at work will have a positive impact [11].

A leadership that ensures that desirable and undesirable work tasks are distributed fairly and ensuring sufficient resources is more likely to result in academics developing a social community at work as there may be less infighting over resources.

Our third and final hypothesis thus reads:

**Hypothesis** **3** **(H3).***High administrative resources buffer the negative impact of unfair leaders on the development of a sense of community*.

## 2. Materials and Methods

### 2.1. Study Design

The survey data set consisted of *n* =1499 academics employed at 10 different Norwegian universities and university colleges. The respondents were all part of universities and university colleges participating in the ARK intervention programme (Norwegian acronym for “Working environment and working climate surveys”). The sample reflected the Norwegian population quite well in that it included both larger and smaller universities and university colleges with a wide range of faculties and departments. The universities and university colleges with their different departments were anonymized due to ethical consideration and therefore we have no information regarding response rate. ARK is a comprehensive research-based plan and tool for systematic mapping of the psychosocial work environment and development and implementation of interventions for improving well-being, health, and performance [38]. The study had a cross sectional design. The data were collected in the period March 2015–February 2020. Due to concerns of anonymity, the data set did not provide any identification of which faculties and departments the respondents belonged to. The ARK programme is an ongoing systematic programme for promoting the work environment and climate in Norwegian Universities and university colleges. The programme has been running since 2012, and all the data are gathered in a common database that can be used for research. The variables chosen here was included in 2015 and, therefore, we used all the data that had been gathered until 2020.

The ARK Intervention Program sent e-mails to the participants containing a link to the KIWEST (Knowledge Intensive Working Environment Survey Target) survey. The e-mail also informed that their participation was voluntary, and that the data would be treated with confidentiality. In addition, they were informed that the project was reported to the Data Protection Official for Research, Norwegian Social Science Data Services A/S that anonymized data could be used for research purposes, and that approval from the Norwegian Data Protection Authority had been obtained. On the start page of the survey, the participants were informed on how to give and withhold consent. Based on this the ethical standards were satisfied.

### 2.2. Participants

The sample comprised 55% males and 45% females showing that it was evenly distributed. The age distribution was as follows: 8% were under 30 years, 20% were in the category 30–39 years, 28% were in the category 40–49, 25% were in the category 50–59, and 19% were 60 years or more. Furthermore, the majority of the sample, 85%, had a position as tenured professor or associate professors while 15% had a position as doctoral research fellow. In terms of employment, most of the sample, 74%, had a permanent position and 26% a temporary position. The majority, 89%, had a full position.

### 2.3. Measures

The survey data was collected using the mapping tool KIWEST, which is the survey tool used in the ARK Intervention Program [28]. KIWEST was designed to assess the psychosocial factors among employees in the university sector. As an integrated part of the ARK Intervention Program, an important goal in the design of the questionnaire was that it could be used both in interventions at workplaces and in research. Following a Job Demands-Resources framework [12], both job demands and resources in established work environment and work attitude scales were selected on the basis of their reported reliability, validity, and suitability for academic work life. Data were collected digitally by using the survey data collection software SelectSurvey.

Work engagement. Work engagement was assessed using the validated Norwegian short version of the Utrecht Work Engagement Scale-9 (UWES-9) [12,39]. The short version contains nine items that participants rate on a seven-item scale from “never” (1) to “every day” (7). UWES-9 measures three sub-dimensions of work engagement, vigour, dedication and absorption, with three items pertaining to each. An example of an item for vigour is “at my work, I feel bursting with energy”; for dedication “I am enthusiastic about my job”; and for absorption “I feel happy when I am working intensely”. Cronbach’s alpha was 0.93.

Fair leadership. The respondents’ experiences of the leaders´ fairness were measured by one dimension consisting of three items from The General Nordic questionnaire for psychological and social factors at work (QPS-Nordic) [40]. The response alternatives ranged from 1 (strongly disagree) to 5 (strongly agree). The statements were asked related to their immediate supervisor with personnel responsibility. Some minor changes were made regarding the original scale to adapt it for use in the academic context. Regarding the original item “My immediate superior distributes work assignments fairly and impartially”. The word “impartially” was removed due to it being double-barrelled. The same was done with the next item “My immediate superior treats the employees fairly and equally” were the word “Equally” was removed. Finally, a new item replaced the item “Is the relationship between you and your immediate superior a source of stress to you”, to “My immediate treats the employees impartially”. Cronbach’s alpha was 0.88.

Social community at work. Social community at work was assessed with three items validated in the second version of the Copenhagen Psychosocial Questionnaire II [41], with the exception of one item which was replaced. This item measured degree of cooperation (“Is there good cooperation between the colleagues at work?”) with an item measuring degree of fellowship (“There is a good sense of fellowship between the colleagues at my unit”). The reason being that ARK qualitatively investigated the academics’ conception of cooperation, which revealed a competitive climate that they generally did not see as mutually exclusive of a strong sense of social community at work. Thus, the added item provided a focus on a sense of social community more appropriate for an academic context. The participants answered on a five-item scale ranging from “strongly disagree” (1) to “strongly agree” (5). Cronbach’s alpha was 0.85.

Resources for teaching. The single item was developed for KIWEST and the aim was to investigate the respondents experience about available necessary resources for their teaching responsibilities. The response alternatives ranged from 1 (strongly disagree) to 5 (strongly agree), The item was “I get the administrative support I need for planning and implementation of teaching and examinations”.

### 2.4. Statistical Analysis

Analysis was conducted on the 1499 individuals who were either in full academic roles or were doctoral research fellows, and who responded to the question on academic resources scale.

Hypothesis 1 was tested with regression analysis. Hypotheses 2 and 3 were tested using the PROCESS macro [42] to examine the indirect (mediator) effect (Hypothesis 2–PROCESS model 4), and the moderated indirect effect (Hypothesis 3–PROCESS model 7), with indirect effects tested using bootstrapping.

All analysis controlled for the age, sex, and academic role of the respondent.

## 3. Results

Table 1 shows the means, standard deviations, and intercorrelations of all study variables (including the control variables). It is noteworthy that the sample is quite evenly distributed between men and women (55% are male), and a large majority (85%) were full academics, rather than doctoral research fellows.

Table 2 shows results of the regression analysis to test Hypothesis 1. It can be seen that, after controlling for age, sex, and academic position, academic resources is moderately and positively related to engagement (B = 0.166, 95% CI (0.123, 0.210), *p* < 0.001). Therefore Hypothesis 1 is supported.

Table 3 shows results of the constituent regression analyses to test the indirect effect proposed by Hypothesis 2. The indirect effect itself was found to be 0.072, with a bootstrap 95% confidence interval of (0.055, 0.095), which clearly does not include zero. Therefore Hypothesis 2 is supported. The ratio of the indirect to total effect (P_m_) [43] was 0.433, indicating that 43.3% of the relationship between academic resources and engagement could be explained by social community at work.

Table 4 shows results of the constituent regression analyses to test the moderated indirect effects proposed by Hypothesis 3 (with fair leadership as moderator). It can be seen that the interaction between academic resources and fair leadership significantly predicts social community; the negative coefficient indicating that the positive relationship between academic resources and social community is reduced when leadership is fairer.

This is indicated by the plot in Figure 1: this shows that the negative effect of less fair leadership is partly mitigated by having better resources. The moderated indirect effect was significant; the index of moderated mediation was −0.017 (bootstrap 95% confidence interval of (−0.033, −0.003)), and the indirect effect itself was stronger (more positive) when leadership was less fair (indirect effect of 0.045), than when leadership was more fair (indirect effect of 0.015). Therefore Hypothesis 3 was supported.

## 4. Discussion

In the present study, we set out to explore the positive aspects of academic working life. In light of decades of research focusing on the negative aspects of working in academia including the lack of administrative resources available to support teaching duties, we proposed that having sufficient resources may be an important factor in academics’ work engagement. Previous research has found collegiality to be an important characteristics of high-performing research teams [44], and work engagement has been found to be related to performance [7]. Therefore, it is reasonable to assume that creating positive work environments characterized by sufficient academic resources may not only benefit academics themselves but also universities as a whole as healthy, engaged workers are also more productive [7,45]. There are of course other important resources for academics as well. However, in the context of the academic sector with increasing demands and lack of resources it was of great importance to investigate the role of administrative resources. Together our findings add to the emerging body of research on administrative resources in the workplace and the mechanism and contextual factors that may help explain when and how such resource is associated with work engagement. Given the adverse market forces in academia, it is important to study how we may strengthen work engagement. Our study points to important factors that management can ensure are in place to promote work engagement: making adequate administrative resource available and ensuring that such resources are fairly distributed by leaders.

We replicated the findings of [20] that sufficient administrative resources to support teaching duties were positively associated with work engagement. We found that a sense of community mediated the relationship between academic resource for teaching and work engagement. This result suggests that one way to overcome the competitive environment in academia is to build academics’ social resources by providing them with the necessary resources to perform a core part of their jobs, namely teaching and examining students. The relationship between academic resources and a social community at work was moderated by fair leadership. Using COR theory [23] and passageways as a theoretical framework, we suggest that fair leadership may be an important contextual factor, which may enhance the positive association between having sufficient administrative resources and experiencing a sense of community with one’s colleagues. We propose that where leadership is fair this may make the impact of adequate resource even stronger as fair leadership may mean that these resources are fairly distributed among staff.

### 4.1. Implications for Practice

The psychosocial work environment needs to be put on the agenda also in universities and for leaders in the academic setting. Leaders need to set aside time and resources to work systematically on maintaining and increasing job resources and to develop administrative resources in order to help build a positive climate for a sense of fairness, collaboration and social community and, thereby, work engagement. Our results indicate that much can be gained by ensuring academics have the necessary resources to perform their teaching duties. Teaching duties are often valued as less valuable for promotion and carrier development than research obligations in academia. To have sufficient resources to deal with the teaching obligations will create conditions that will ease the workloads regarding teaching, help academics balance the relationship between teaching and research, and contribute to build a more collaborative and positive community. Leaders in academia are selected on the basis of their research skills not their leadership skills and rarely undergo sufficient training to perform this role. It would appear that creating an environment where resources enable academics to perform their teaching responsibilities may protect against poor leadership. Leadership in academia has much to do with creating conditions for the employees to conduct their research and teaching. For the leaders to promote organizational justice, fairness and social community one important key word is communication. Fair leadership is demonstrated and communicated through open and transparent processes where distribution of resources and measures for improving the resource situation becomes visible. Furthermore, it is crucial to create shared goals and create meeting places and time for both formal and informal communication and meetings.

### 4.2. Strengths and Limitations

The main strength of our study is the large sample size, however, our study also suffers some limitations which must be considered.

First, in line with Han and colleagues [20] we focused on administrative support for teaching responsibilities of academics. However, academics also have major research responsibilities and future studies should study the importance of administrative support for other key academic responsibilities.

Second, also in line with Han and colleagues [20], we employed a cross-sectional study design. The cross-sectional design prevents us from drawing causal conclusions and our results should, therefore, be interpreted with caution. Future studies should employ a longitudinal study design.

Finally, we measured administrative resources using a single item measure. This is a potential limitation as it can lead to lower reliability and, therefore, lower statistical power. It may also raise questions about the validity of the measure. Having said that, the use of single item measures is not unusual and may have higher face validity and better convergent validity [46]. Further research would benefit from including questions regarding, administrative and technical/digital support for research and research applications, internationalization and dissemination.

## 5. Conclusions

Increased workload, work group conflicts, competitive environments and challenges regarding balancing research and teaching put an emphasis on the importance of the role of leadership and developing a good and resourceful psychosocial work environment for academics. Motivated and engaged employees are important for meeting high quality standards in both research and teaching in universities today, and therefore we investigated factors that may promote work engagement among academics in universities and university colleges in Norway through the lens of a positive psychology. A significant contribution of this study is the examination of the positive relationship between administrative resources to fulfil teaching responsibilities and work engagement, and furthermore the examination of how a sense of community may explain this link. The relationship between academic resources and a social community at work is moderated by fair leadership meaning that administrative resources may also buffer the impact of a leadership that is perceived to be unfair. Aiming for a positive future for both academics and the universities in themselves, it is crucial to put the psychosocial work environment on the agenda and through a positive psychological focus build both administrative and job-related resources.

## Figures and Tables

**Figure 1 ijerph-18-02776-f001:**
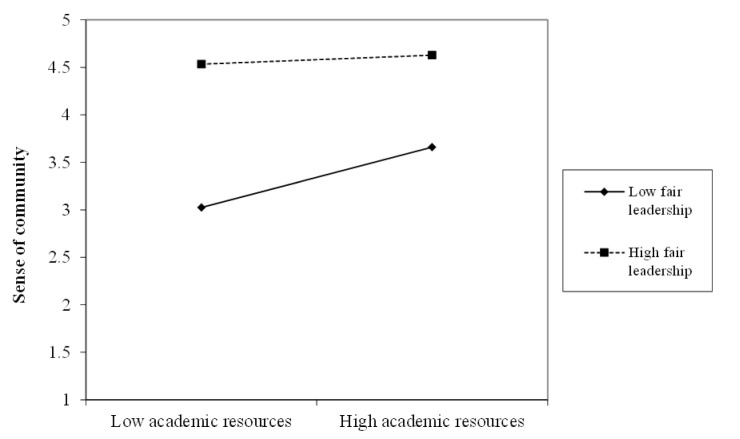
Interaction between academic resources and fair leadership predicting social community at work.

**Table 1 ijerph-18-02776-t001:** Descriptive statistics and intercorrelations of study variables.

Study Variables	Mean	SD	Correlations						
			1	2	3	4	5	6	7
1. Academic resources	3.56	1.08							
2. Social community at work	3.95	0.77	0.27 ***						
3. Engagement	4.71	0.94	0.18 ***	0.33 ***					
4. Fair leadership	3.73	0.89	0.34 ***	0.49 ***	0.23 ***				
5. Age ^1^			−0.08 **	−0.04	0.07 **	−0.10 ***	−0.13 ***		
6. Sex ^2^			0.08 **	0.00	−0.07 **	0.07 **	−0.02	0.08 **	
7. Academic position ^3^			−0.04	0.01	0.07 **	−0.06 *	−0.08 **	0.53 ***	0.06 *

^1^ Age: 1 = under 30; 2 = 30–39; 3 = 40–49; 4 = 50–59; 5 = 60 or over. ^2^ Male = 1, Female = 0. ^3^ Full academic = 1, Doctoral research fellow = 0. Correlations of at least 0.05 have *p* < 0.05 *; correlations of at least 0.07 have *p* < 0.01 **; and correlations of at least 0.09 have *p* < 0.001 ***.

**Table 2 ijerph-18-02776-t002:** Results of regression analysis testing Hypothesis 1.

Dependent Variable	Engagement	
	B (95% Confidence Interval, CI)	*p*
Age	0.054 (0.008, 0.100)	0.022
Sex	−0.172 (−0.267, −0.078)	0.000
Academic position	−0.110 (−0.265, 0.045)	0.164
Academic resources	0.166 (0.123, 0.210)	0.000
R^2^	0.048	

Figures in main section of table are unstandardized regression (B) coefficients.

**Table 3 ijerph-18-02776-t003:** Results of regression analysis testing Hypothesis 2.

Dependent Variable	Social Community at Work		Engagement	
	B (95% CI)	*p*	B (95% CI)	*p*
Age	−0.022 (−0.059, 0.016)	0.256	0.062 (0.18, 0.106)	0.006
Sex	−0.035 (−0.112, 0.041)	0.364	−0.159 (−0.250, −0.069)	0.001
Academic position	−0.079 (−0.204, 0.046)	0.214	−0.081 (−0.228, 0.067)	0.286
Academic resources	0.194 (0.259, 0.229)	0.000	0.094 (0.052, 0.137)	0.000
Social community at work			0.372 (0.311, 0.431)	0.000
R^2^	0.075		0.134	

Figures in main section of table are unstandardized regression (B) coefficients.

**Table 4 ijerph-18-02776-t004:** Results of regression analysis testing Hypothesis 3

Dependent Variable	Social Community at Work		Engagement	
	B (95% CI)	*p*	B (95% CI)	*p*
Age	0.002 (−0.032, 0.035)	0.927	0.062 (0.18, 0.106)	0.006
Sex	−0.069 (−0.137, −0.001)	0.047	−0.159 (−0.250, −0.069)	0.001
Academic position	−0.095 (−0.206, 0.017)	0.097	−0.081 (−0.228, 0.067)	0.286
Academic resources	0.249 (0.133, 0.364)	0.000	0.094 (0.052, 0.137)	0.000
Social community at work			0.372 (0.311, 0.431)	0.000
Fair leadership	0.548 (0.438, 0.658)	0.000		
Academic resources × fair leadership	−0.045 (−0.076, −0.014)	0.004		
R^2^	0.075		0.134	

Figures in main section of table are unstandardized regression (B) coefficients.

## Data Availability

The data can be found and applied for at the ARK research platform at the HUNT Databank: ARK—Arbeidsmiljø og miljøundersøkelser (ntnu.no).

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
