# Peer review of "The Role of Adequate Resources, Community and Supportive Leadership in Creating Engaged Academics"

_ijerph, 2021, doi:10.3390/ijerph18052776_

Round 1
Reviewer 1 Report
Dear authors,
Thanks for your responses and the review of each section in the manuscript. You have addressed all the points of the previous revision.
I suggest that you look at few typos in the manuscript, for example in pag.3
Previous research has found the research has found fairness and 105 reasonable job responsibilities are related to sense of community
In addition, in Table 1 and 3, the word “Dependent Variables” is located on top of the list of Independent variables, instead of being on top of Social Community and Work and Work Engagement (the dependent variables in the model). This might be confusing.
Thanks for your contribution!
Reviewer 2 Report
The revised version appears to me to be acceptable for publication in this journal. Edits appear to have strengthened the manuscript.
Reviewer 3 Report
I think the authors have improved the paper considerably. In particular, the introduction has improved a lot, so has the presentation of the results, the discussion is more focused on the paper's results, and limitations reflect better the limitations of the paper.
My main concerns remain about the validity of the measure 'administrative resources' (one-item measure) as the independent variable, and the very low R2 explained in the model. Both imply the practical relevance of these results is very limited.
As minor comments:
(1) the term academic resources still appears most times throughout the text and in the tables; it should be changed most times for administrative resources as the authors say to have done (but many have not).
(2) In line 70 they state 'they replicate the results by Han et al. (2020), whereas Han et al. consider a wide range of resources and demands. They should rewrite which part of Han they replicate.
(3) minor errors (Hypothesis 3a in Table title 4), and a good revision of the many mistakes in English grammar and spelling needs to be done.
This manuscript is a resubmission of an earlier submission. The following is a list of the peer review reports and author responses from that submission.
Round 1
Reviewer 1 Report
I would like to thank the authors for their effort in developing this Manuscript: “The Role of Adequate Resources, Community and Supportive Leadership in Creating Engaged Academics”. It is an interesting article with important implications to create psychologically healthier academic institutions. In the following section, I provide some comments and suggestions to the authors to consider the previous publication of this manuscript.
Introduction
In the Hypothesis development (p. 2; line 49), the authors indicate that in the present study, “the goal is work engagement”. It not clear if this was an actual goal of the participants, or if the “goal” refers to a conceptual attribute of work engagement within the COR theory to justify its inclusion in the hypothesized model. More clarity is needed.
The conceptualization of academic resources is too broad, and it is not clear in the introduction what specific resources are included. It is not clear if it refers to administrative support (on page 2 line 50, administrative support is discussed). I would recommend the authors to clarify the concept of “academic resource” and to tie its measure.
More details should be provided in the explanation of the mediating role of social community at work.
Method
In the hypothesis, the authors indicate that the measurements were collected at different moments in time (T1 & T2), however, in the methods section (p.3 line 127-129) the authors indicate that this is a cross-sectional design and an anonymous survey. If the survey was anonymous and cross-sectional, the hypothesis should be reviewed to eliminate the T1 & T2.
“The study had a cross-sectional design. The data was collected in the time period of March 2015 – February 2020. Due to concerns of anonymity, the data set did not provide any identification of which faculties and departments the respondents belonged to.”
In terms of the demographic variables, is there any difference between those holding a permanent and a temporary position in the study variables? Will the model be equivalent to those two groups? Are working conditions/ benefits similar?
In table 1, Age, Sex and Academic position have Mean and SD, even though these are categorical variables. I suggest eliminating these estimates. Additionally, no correlation is marked as significant or not significant. There is a note in the table, but no asterisk or other mark is used to identify the significant correlations.
Discussion
The authors present an interesting finding related to the moderating effect of fair leadership; however, the discussion of this result needs to be elaborated. Based on the COR theory, how can this result be explained? What other explanations could be provided regarding the role of fair leadership in academia?
How are these results important in the promotion of work engagement? Nothing is said about the dependent variable in the discussion.
More specific implications to practice need to be provided. How can academia support academic resources and promote collegiality a sense of community? What specific practices should be implemented? If fair leaderships seem to buffer the positive effect between resources and a sense of community, how can we promote organizational justice and fairness in academia? What can leaders do to prevent such a negative effect?
In terms of the limitation, I would suggest the authors consider that they only include one resource (also measure with a single item). Additional academic resources should be included in future studies, which other resources may be relevant?
Conclusion.
It is not clear how this conclusion is based on the results and discussion provided: “Being able to access such resources may also buffer the impact of a leadership that is perceived to be unfair.” Please, elaborate.
Reviewer 2 Report
With such good research, it would be necessary to expand the conclusions.
Reviewer 3 Report
Thank you for the opportunity to review this manuscript. It takes on an important topic and in a novel way. The theoretical frame also seems strong and well chosen. In my view, this manuscript is in strong condition overall. Here are just a small number of suggestions:
- re: the survey instrument - please share info about how reflective the sample is of the population of interest (or explain if not known)
- re: the survey instrument - please share info re: response rate (if known)
- re: the survey instrument - why the long time window (2015-2020), and are there any consequences that this introduces (eg limitations, challenges with interpretation)
- it seems participants are asked to think about their supervisor's leadership in particular. If a larger system of leadership/supports exists at these institution, does this introduce a limitation or a challenge in terms of interpreting 'fair leadership' as part of this study? in any case, authors might want to address this more explicitly in terms of methods or discussion
Reviewer 4 Report
The manuscript about ‘The Role of Adequate Resources, Community and Supportive Leadership in Creating Engaged Academics’ covers an important topic, improving academics working conditions in an increasingly competitive and unhealthy environment. It does so from the perspective of Positive Psychology putting the focus on resources which could help improve the current situation and improve engagement and performance in Academia.
Yet, the manuscript has currently important limitations. I will raise some of them in trying to help you improve the manuscript.
- The topic is relevant and is well grounded in appropriate theoretical models (COR, JDR, etc). In my opinion, the weakest point is the very restricted variable choice in the empirical model which does not meet the expectations set in the abstract and introduction. In this line, ‘academic resources ‘ are restricted to one item regarding administrative support for teaching and examinations (on this item, I am not quite sure what this means, making photocopies and introducing the marks in the students files?). ‘Social resources ‘are restricted to fair leadership and social community (which then is measured regarding the distribution of work and interpersonal justice; social community measured as a sense of community rather than a strong social support context to create engaged work). The research model is then based on these three resources and their relationships. These three resources are too limited in the world of resources needed to develop teaching and researching in Academia, and even more so to affect the level of engagement and performance.
I suggest the authors enrich this model with further resources they may find in the KIWEST survey. Besides traditionally relevant resources which have not been considered (e.g. autonomy), instrumental support in the form of knowledge sharing among colleagues (about theory, research skills, statistical analysis, writing papers, developing and presenting research projects), all these types of support may be more directly related to engagement and performance than ‘administrative support’ and ‘fair distribution by the leadership’. Most likely with a wide survey of such big sample there is data available for the authors to enrich this model, and more so if data have been collected at different point times.
- The relationships between constructs and rationale to support these relationships are weak too. For instance, in lines 67-82, the authors present a bunch of concepts on resource gain spirals, a competitive environment, performance metrics, even neo-liberal thinking, and the relevance of social resources. While it may be true that ‘a social community at work may be an important mediator of the relationship between academic resources and work engagement’, there is a big step between this statement and the final proposal that “administrative support creates a sense of community among academics which then leads to engagement”. This forcing of big arguments into narrow variables happens a few times in the paper. Further, the data in table 2 and 3 shows that the actual impact of academic resources both on engagement (R2=.048) and Social community (.075) is really low. There is actually very littel variance explained. What is the practical significance of this model and results?
- At some points the writing of the manuscript seems to have been rushed. There are some misspellings (line 66 theoretical perceptive; line 185 recourses) and some sentences difficult to follow, but more importantly the hypotheses have been stated with two times (T1, T2) that seem to disappear in the manuscript; the results section includes data on ‘empowering leadership’ (Table 1, Table 5). There is a mention in lines 114-116 to empowering leadership but then this concept disappear again.
- The study design says the collection period ranged from 2015-2020. Please explain how this big time range may affect the results, as the reality to which academics answers may have changed across these years. Further, it would be interesting to know how the sample distribution relates to the actual population of university teachers. Further on the method, there are some relevant weaknesses, the independent variable is based on one-item, the design is cross-sectional but it seems there may be longitudinal data available.
- The arguments in the discussion are again ‘grand’ compared to the actual contribution of the empirical study: ‘We set out to explore the positive aspects of academic working life’ (some restricted positive aspects); “creating positive work environments characterized by sufficient academic resources” (I would question that administrative aid may be called ‘sufficient academic resources’).
- In lines 271-276, the authors say they found support for mediated moderation but contrary to expectations. I think the explanations given by the authors could improve in clarity both when the hypothesis is developed and the results interpreted. Looking at Figure 1, I am interpreting that the interaction is in the right direction. You can actually put the focus of the moderation on either leadership or academic resources. Maybe, you could state that high academic resources buffer the negative impact of unfair leaders on the development of a sense of community. Is that contrary to expectations?
Final comment, the topic you present is interesting and of practical relevance for academics working conditions. I hope you can improve your paper by adding relevant variables from your survey and wish you good luck with the endeavour.